# One in five South Africans are multimorbid: An analysis of the 2016 demographic and health survey

**Rifqah Abeeda Roomaney**[1,2]*, **Brian van Wyk**[2], **Annibale Cois**[1,3], **Victoria Pillay-van Wyk**[1]

**1** Burden of Disease Research Unit, South African Medical Research Council, Cape Town, Western Cape, South Africa, **2** School of Public Health, University of the Western Cape, Cape Town, Western Cape, South Africa, **3** Division of Health Systems and Public Health, Department of Global Health, University of Stellenbosch, Western Cape South Africa

* rifqah.roomaney@mrc.ac.za

**Data Availability Statement:** The third party data underlying the results presented in the study are available from The DHS Program. Users can register on The DHS Program website (https://dhsprogram.com/). Once registered, interested researchers can request access to the DHS

## Abstract

Multimorbidity is a global research priority, yet relatively little is known about it in low and middle income countries. South Africa has the largest burden of HIV worldwide but also has a growing burden of non-communicable diseases; potentially leading to uncommon disease combinations. Information about the prevalence of multimorbidity and factors associated with it can assist in healthcare planning and targeting groups of people for interventions. This study aimed to determine the prevalence of multimorbidity by age and sex, as well as factors associated with multimorbidity in people 15 years and older. This study analyses the nationally representative 2016 South African Demographic Health Survey. The sample included 10 336 people who participated in the Adult Health questionnaire and approximately 7 961 people who provided biomarkers. Multivariate logistic regression was used to measure the association of multimorbidity with age, sex, living in an urban or rural area, education level, wealth level, employment status, body mass index, current alcohol or tobacco use. All analyses were conducted using STATA 15. Multimorbidity was present in 20.7% (95% CI: 19.5%– 21.9%) of participants; in 14.8% (95% CI: 13.4% - 16.3%) of males and 26.2% (95% CI: 24.7–27.7%) of females. Multimorbidity increased with age; with the highest odds in the 55–64 years old age group (OR: 24.910, 95% CI: 14.901–41.641, p < 0.001) compared to those aged 15–24 years. The odds of multimorbidity was also higher in young females compared to young males (OR: 2.734, 95% CI: 1.50–4.99, p = 0.001). Possessing tertiary education (OR: 0.722, 95% CI: 0.537–0.97, p = 0.031), being employed (OR: 0.813, 95% CI: 0.675–0.979, p = 0.029) or currently using alcohol (OR: 0.815, 95% CI: 0.686– 0.968, p = 0.02) was protective against multimorbidity. Multimorbidity is prevalent within the South African population, with females and older adults being most affected. However, multimorbidity is also observed in younger adults and most likely driven by the high prevalence of HIV and hypertension.

datasets. The 2016 South African DHS data is available for download at the following link: (https://dhsprogram.com/data/dataset/South-Africa_Standard-DHS_2016.cfm?flag=0). The authors confirm they had no special access privileges.

**Funding:** The work reported herein was made possible through funding by the Burden of Disease Research Unit at the South African Medical Research Council. RAR conducted this research under the South African Medical Research Council through its Division of Research Capacity Development under the Internship Scholarship Programme from funding received from the South African National Treasury. The content hereof is the sole responsibility of the authors and does not necessarily represent the official views of the South African Medical Research Council or the funders. Grant number: NA.

**Competing interests:** The authors have declared that no competing interests exist.

## Introduction

People living with more than one disease (also known as multimorbidity) have their lives impacted in many ways; including a reduced quality of life [1–6], an increased risk of dying [7–9] and an intensified need to utilise healthcare [10–14]. Despite these negative impacts, the area of multimorbidity remains under researched when compared to research afforded to single disease conditions [15]. This is particularly acute in low and middle income countries (LMICs) where 5% of multimorbidity research globally has taken place [15]. Little is known about multimorbidity in LMICs where disease burdens are thought to differ from countries with more established multimorbidity profiles.

South Africa is an upper middle-income country [16] with a quadruple burden of disease consisting of: HIV/AIDS and tuberculosis (TB); other communicable diseases, perinatal conditions, maternal causes, and nutritional deficiencies; non-communicable diseases (NCDs); and injuries [17]. South Africa has a very high HIV prevalence and it is not uncommon that people living with HIV also develop other chronic conditions. Given this HIV burden, information is needed on the prevalence of multimorbidity to plan for more responsive healthcare services. This information is valuable for planning purposes as health service delivery could be made more efficient around common disease clusters to the benefit of those living with multimorbidity. Knowing who is most affected by multimorbidity (i.e. determinants or factors that are common in those affected) can also be used to design interventions to target those individuals. While multimorbidity research has been emerging in the country for the past decade, few studies have reported the prevalence of multimorbidity and factors associated with it in a consistent and comparable manner [18]. The authors conducted a systematic review of multimorbidity prevalence studies in South Africa and found significant heterogeneity in the study designs as well as the estimates of prevalence [18]. Of the studies included [19–27], the prevalence of multimorbidity ranged from 3 to 87%. In addition, the factors associated with multimorbidity were disparate and at times contradictory. Among the factors that were occasionally associated with multimorbidity in South Africa were: age, being female, locality, education level, body mass index (BMI) and marital status.

Prevalence estimates form an important part of the information used for evidence-based health decision-making. Given the lack of studies conducted about multimorbidity in South Africa, we aimed to determine the prevalence of multimorbidity by age group and sex in the country using the 2016 Demographic and Health Surveys (DHS) 6 (SADHS 2016). In addition, this paper reports the process and results derived from a systematic analysis of the SADHS 2016 to establish factors associated with multimorbidity in the South African population. The SADHS 2016 is unique in South Africa in that it is a nationally representative survey which includes biomarkers for the measurement of HIV, HbA1c (diabetes), blood pressure and anaemia status.

## Materials and methods

### Sample and data source

National survey data is an important source of information about multimorbidity. National surveys represent a largely untapped resource that could shed light on multimorbidity in the general population. This is especially true for LMICs such as South Africa where limited information exists about multimorbidity. The DHS project, primarily funded by the United States Agency for International Development has conducted more than 230 nationally representative comparable household surveys in more than 80 countries since 1984 [28]. The DHS collects data on a range of topics such as fertility, contraception, maternal and child health, HIV,

malaria and domestic violence [28]. For many countries, the DHS is an important source of information for policy making, monitoring and evaluation and as the country's public health evidence base [28]. In terms of multimorbidity, the DHS collects information on self-reported health conditions and biological markers.

This paper presents a secondary analysis of national survey data from the SADHS 2016. The survey is nationally representative with the aim of providing up-to-date estimates of demographic and health indicators such as information on fertility levels, marriage, sexual activity, contraceptive use, nutrition, child mortality, aspects of child health, exposure to the risk of HIV infection, behaviour and health indicators [29]. The SADHS 2016 also collected information on anthropometry, anaemia, hypertension, HbA1c levels and HIV among adults 15 years and older.

The SADHS 2016 followed a stratified two-stage sample design and a total of 750 primary sampling units (PSUs) were selected and stratified by urban, traditional and farm areas. A fixed number of twenty dwelling units were randomly selected in each PSU. Of these dwelling units, sub-sampling occurred whereby half of the households were eligible for a South African-specific module on Adult Health that included the collection of biomarkers [29].

All participants signed consent forms to participate in the study SADHS 2016. For this secondary data analysis, the anonymised dataset with necessary permissions was obtained from the DHS programme. In addition, ethics clearance was granted by the Biomedical Research Ethics Committee of the University of the Western Cape (BM20/5/8) as part of the lead author's PhD project.

## Description of included variables

Multimorbidity is frequently measured by counting the number of co-existing conditions, using a predefined list of medical conditions [30, 31]. Various studies have used this technique when doing secondary data analysis [32]. Estimation of multimorbidity included: self-reported diseases (e.g. bronchitis/ COPD, heart disease, high blood cholesterol, stroke, TB in the last 12 months), biomarker disease (e.g. HIV, anaemia, high blood pressure) and a combination of the two (i.e. diabetes). Disease variables were coded as binary (disease absent '0' or disease present '1'). An index variable was created where for each individual, the number of disease conditions present was counted. If there was information about a disease condition missing, this was counted as "no disease present". The disease index variable was further categorised to create another variable, the Multimorbidity Index. This variable categorised individuals into either having "no multimorbidity" (no disease or only one disease present) or "multimorbidity present" (two diseases or more present).

**Self-reported diseases.** The study sample consisted of 10 336 youth and adults who completed the Adult Health module and were asked about the presence of several diseases (Table 1) e.g. "Has a doctor, nurse or health worker told you that you have or have had any of the following conditions". The response to the questions were "No", "Yes" or "Don't know", with the "Don't know" response recorded as missing values. TB in the last 12 months was constructed from two other variables: whether a participant had ever had the disease and whether they had the disease in the last 12 months or more than 12 months ago.

The analysis included data where the variables were deemed to be "current" conditions. Disease conditions were excluded for the following reasons: (i) disease conditions that could not be assumed to be current at the time of the survey due to the way that the question was asked (ii) disease conditions that were considered to be acute or of short duration (iii) disabilities or injuries. Two clinicians assisted where the information was unclear. Further details are available in S1 Table in S1 File.

**Table 1. Survey questions on self-reported diseases.**

| Variable | Survey Question |
|---|---|
| **Diabetes** | Has a doctor, nurse or health worker told you that you have or have had any of the following conditions: diabetes or blood sugar? |
| **Emphysema/ Bronchitis/ COPD** | Has a doctor, nurse or health worker told you that you have or have had any of the following conditions: chronic bronchitis, emphysema, or COPD? |
| **Heart disease** | Has a doctor, nurse or health worker told you that you have or have had any of the following conditions: Heart attack or angina/chest pains? |
| **High blood cholesterol** | Has a doctor, nurse or health worker told you that you have or have had any of the following conditions: high blood cholesterol or fats in the blood? |
| **Stroke** | Has a doctor, nurse or health worker told you that you have or have had any of the following conditions: stroke? |
| **TB in the last 12 months** | Has a doctor, nurse or health worker ever told you that you have TB? |
|  | When was the last time you had TB? |

**Physically measured diseases (biomarkers).** Of the people included, approximately 74.4% (n = 7 961) of people also had information on physically measured diseases. The following information was of interest to the analysis: diabetes (HbA1c), HIV status (dry blood spot), blood pressure measurements, anaemia (Hb), anthropometry (height and weight). Nurses collected blood specimens from finger pricks.

For diabetes, dry blood spots were analysed using a blood chemistry analyser which measures total haemoglobin concentrations [29]. A participant was assigned diabetic status if their HbA1c $\geq$ 6.5 mmol [33, 34]. Participants with normal HbA1c values but on medication to manage diabetes were also assigned diabetic status. For participants without HbA1c data, their disease status was based on their self-assessment of whether they had diabetes or not.

For HIV, dry blood spots were tested with an enzyme-linked immunosorbent assay (ELISA) and a second ELISA was done for confirmation [29]. The results of the first ELISA was included in this study.

For anaemia, nurses collected blood samples in a microcuvette and the analysis of haemoglobin was conducted on site. The SADHS 2016 anaemia results were adjusted for smoking status and altitude [29]. Anaemia levels below 7.0 g/dl were considered as severe anaemia. Moderate anaemia was considered levels between 7.0g/dl and 9.9g/dl. For pregnant women, mild anaemia were levels between 10.0 g/dl and 10.9 g/dl and between 10.0 g/dl and 11.9 g/dl for all other adult women [35]. Participants were then categorized either having no anaemia or having anaemia. The degree of anaemia was characterized as mild, moderate or severe.

Three blood pressure measurements were taken from participants using digital blood pressure monitors [29]. For this study, the first measurement was excluded and the average of the remaining repeated measurements were taken. The values were categorised as: hypertension absent (Systolic < 120 mmHg & diastolic < 80 mmHg), Pre-hypertension (Systolic: 120–139 *mmHg* or diastolic: 80–89 *mmHg*), Stage 1 Hypertension: (Systolic: 140–159 *mmHg* or diastolic: 90–99 *mmHg*), Stage 2 hypertension (Systolic $\geq$160 *mmHg* or diastolic $\geq$100 *mmHg*) [36]. Hypertension was coded as being absent (normal or pre-hypertension) or present (Stage 1 or Stage 2 hypertension). People on medication to manage hypertension were included in those that had hypertension. Data cleaning for diabetes [37] and hypertension [38] followed the procedures used in the Second South African Comparative Risk Assessment. Further details of data collection, cleaning and coding is listed in S2 Table in S1 File.

**Other variables of interest.** Systematic reviews identified the following characteristics (among others) as being related to multimorbidity: (i) Biomedical and individual: ageing, female, (ii) Socioeconomic: lower socioeconomic status, high-income group (in low and

middle-income countries), lower education, (iii) Social and environmental: living in urban environments (iv) Behavioural: tobacco, overweight and obese [39]. For this study, the following variables were investigated as predictor variables: age category, sex, locality, highest education level, wealth index, employment status, BMI category, current smoker status and current alcohol drinker status.

The ages of participants were taken from DHS 2016 dataset. Participants under the age of 15 years were excluded. Where appropriate, age was analysed in 10-year age bands. The variable sex was included and participants were coded as male or female. Locality was included and coded as either urban or rural. Educational attainment was also included and described by the 'highest grade or form you completed at that level'. This study divided the responses into three categories: primary school or less, secondary school, and tertiary education. Employment status was coded as employed (currently working) or unemployed.

This study made use of the SADHS 2016 wealth index. The wealth index uses principal component analysis to score households according to the types of goods that are owned and other characteristics [29]. The households were divided into five quintiles, from poorest (Quintile 1) to richest (Quintile 5).

This study also examined current alcohol and tobacco use. For current alcohol use, the responses to the following two questions were combined: *"Have you ever consumed a drink that contains alcohol such as beer, wine, ciders, spirits, or sorghum beer?"* and *"Was this within the last 12 months?"*. For tobacco use, the question *"Do you currently smoke tobacco every day, some days, or not at all?"*. Both variables were coded as binary (e.g. Yes/No).

The BMI of participants were also examined. Height and weight were measured using a digital scale and stadiometer [29]. BMI was calculated using the *BMI* STATA package. BMI was categorized as follows: underweight (15 .0 - $<18.5$ kg/m$^2$), normal weight (18.5 - $<25.0$ kg/m$^2$), overweight (25.0 - $<30.0$ kg/m$^2$), obesity grade 1 (30.0 - $<35.0$ kg/m$^2$), obesity grade 2 (35.0 - $<40.0$ kg/m$^2$), obesity grade 3 (40.0 - $<60.0$ kg/m$^2$) [40]. Data cleaning was done in accordance to another study [40]. Further details are listed in S2 Table in S1 File.

## Analysis

The statistical analysis was done using STATA 15.0 (Stata Corporation, College Station, Texas, USA) software. The STATA survey set ('*svy*') of commands were used to account for the complex survey design. Sampling weights were calibrated against the Statistics South Africa mid-year population estimates [41].

For unweighted data (sample), frequencies were used to display categorical data. Age was analysed as a continuous variable while gender, locality, province, educational level and wealth index were analysed as categorical variables. Bivariate associations between locality, province, highest education level and wealth index by sex were assessed using Chi-square tests. The prevalence of having single disease conditions by sex was also assessed with Chi-square tests. For weighted data, multimorbidity status was described using histograms and box plots against age.

Regression methods were used to describe the relationship between a dependent variable and other predictor variables [42]. In this case, a multivariate logistic regression was employed because the dependent variable was binary *(Multimorbidity absent = 0, Multimorbidity present = 1)*.

Crude odds ratios were estimated by only including the dependent variable and one predictor variable. Three models were constructed for logistic regression with multimorbidity as the dependent variable.Model 1 contained only demographic information (e.g. age and sex), while Model 2 contained sociodemographic information (e.g. age, sex, educational attainment,

wealth index and employment status). The final model (Model 3) included all variables in the previous models but also included lifestyle or behavioural factors (e.g. alcohol use, tobacco use and BMI).

Model checking was performed using various statistical tests. The link test [43] was used to determine if there were specification errors. Interaction terms were added where necessary. Influential observations were checked using the Pearson residuals, deviance residuals and Pregibon leverage [44] on the unweighted model as these tests cannot be used on survey weighted data. Influential observations were dropped, and the model was refitted. The crude and adjusted odds ratios were reported with 95% CIs and $p$-values of less than 0.05 were considered as statistically significant.

## Results

### Sample description

There were 10 336 youth and adults included in the sample; with more females (59.2%) than males (Table 2). The median age of participants was 36 years (interquartile range: 24–52 years), with females being slightly older than males but this was not statistically significant. More than half of the sample resided in urban areas (55.0%) and most (64.5%) had completed secondary education. The majority of participants were Black African (84.7%), followed by coloured (9.6%), white (4.4%) and Indian/Asian (1.4%). Age, urban location and education did

**Table 2. Description of sample population (unweighted).**

|  | Total (N = 10 336)<br>% (n) | Male (N = 4210)<br>% (n) | Female (N = 6126)<br>% (n) | *p*-value' |
|---|---|---|---|---|
| Age* (Median years and IQR) | 36 (24–52) | 33 (22–49) | 37 (25–54) | 0.442 |
| Urban location | 55.0 (5 685) | 55.2 (2 324) | 54.86 (3 361) | 0.735 |
| Province: |  |  |  | **<0.001** |
| • Western Cape | 7.29 (754) | 6.65 (280) | 7.74 (474) |  |
| • Eastern Cape | 13.08 (1 352) | 13.16 (554) | 13.03 (798) |  |
| • Northern Cape | 8.53 (882) | 8.38 (353) | 8.64 (529) |  |
| • Free State | 9.97 (1 031) | 9.12 (384) | 10.56 (647) |  |
| • Kwa-Zulu Natal | 15.2 (1 571) | 14.32 (603) | 15.8 (968) |  |
| • North West | 10.5 (1 085) | 11.97 (504) | 9.48 (581) |  |
| • Gauteng | 9.97 (1 031) | 11.16 (470) | 9.16 (561) |  |
| • Mpumalanga | 11.8 (1 220) | 12.23 (515) | 11.51 (705) |  |
| • Limpopo | 13.64 (1 410) | 12.99 (547) | 14.09 (863) |  |
| Education level |  |  |  | 0.502 |
| • Primary or less | 26.26 (2 714) | 25.65 (1 080) | 26.67 (1 634) |  |
| • Secondary complete | 64.51 (6 668) | 65.11 (2 741) | 64.1 (3 927) |  |
| • Tertiary | 9.23 (954) | 9.24 (389) | 9.22 (565) |  |
| Wealth index |  |  |  | **0.018** |
| • Quintile 1 (Poorest) | 20.3 (2 098) | 20.45 (861) | 20.19 (1 237) |  |
| • Quintile 2 (Poorer) | 21.55 (2 227) | 22.71 (956) | 20.75 (1 271) |  |
| • Quintile 3 (Middle) | 22.61 (2 337) | 23.06 (971) | 22.3 (1 366) |  |
| • Quintile 4 (Richer) | 19.99 (2 066) | 18.74 (789) | 20.85 (1 277) |  |
| • Quintile 5 (Richest) | 15.56 (1 608) | 15.04 (633) | 15.92 (975) |  |
| Employed | 33.9 (3 506) | 41.6 (1 751) | 28.7 (1 755) | **<0.001** |

*Age in years. 'Categorical variables were tested using Chi-squared, continuous variables tested using Wilcoxon signed rank test.

**Table 3. Prevalence of single disease conditions by sex and method of measurement in South Africa for 2016 (unweighted data).**

| Disease condition | Total % (n/N) | Male % (n/N) | Female % (n/N) | *p*-value |
|---|---|---|---|---|
| **SELF-REPORTED** | | | | |
| Diabetes | 4.5 (459/10 292) | 3.6 (150/4 176) | 5.1 (309/6 116) | <**0.005** |
| Bronchitis/COPD | 1.3 (132/10 290) | 1.0 (40/4 177) | 1.5 (92/6 113) | <**0.005** |
| Heart disease | 3.4 (354/10 305) | 2.4 (101/4 183) | 4.1 (253/6 122) | **0.015** |
| Cholesterol | 2.9 (296/10 282) | 2.3 (94/4 167) | 3.3 (202/6115) | **0.002** |
| Stroke | 1.4 (146/10 309) | 1.0 (40/4 186) | 1.7 (106/6123) | **0.001** |
| TB in last 12 months | 1.3 (138/10 336) | 1.3 (53/4 210) | 1.4 (85/6126) | 0.576 |
| **PHYSICALLY MEASURED (BIOMARKER)** | | | | |
| HIV | 19.9 (1 307/6 584) | 13.8 (346/2 517) | 23.6 (961/4 067) | <**0.005** |
| Hypertension | 46.2 (3 678/7 961) | 45.1 (1 412/3 130) | 46.9 (2 266/4 831) | 0.117 |
| Anaemia | 25.9 (1 862/7 200) | 17.7 (489/2 769) | 31.0 (1 373/4 431) | <**0.001** |
| Diabetes (HbA1c) | 12.4 (839/6 763) | 9.3 (241/2 591) | 14.3 (598/4 172) | <**0.001** |
| **PHYSICALLY MEASURED (BIOMARKER) AND SELF-REPORTED** | | | | |
| Diabetes (self-report or HbA1c) | 10.06 (1 036/10 295) | 7.35 (307/4 178) | 11.92 (729/6 117) | <**0.001** |

not differ between males and females. There were significant differences between the proportion of males and female participants in the sample, by province (p < 0.001) and wealth quintile (p = 0.018).

All self-reported disease conditions were more common in females compared to males (Table 3). Females had a slightly higher prevalence of TB in the last 12 months compared to males, however, the difference was not statistically significant. Other than hypertension, all physically measured disease conditions were significantly more common in females than males. The prevalence of multimorbidity in the sample population was 22.9% (S3 Table in S1 File).

## Prevalence of single diseases and multimorbidity

Table 4 shows the weighted prevalence of each included disease condition by sex. Of the self-reported diseases, diabetes had the highest prevalence (4.4%), followed by high cholesterol (3.5%), heart disease (3.1%), COPD or bronchitis (1.4%), stroke (1.4%) and TB in the last 12 months (1.2%). The prevalence of physically measured diseases was higher than that of self-reported diseases. Of the physically measured diseases, hypertension occurred most frequently (45.0%), followed by anaemia (24.7%), HIV (19.6%) and diabetes (11.7%).

Diabetes was the only disease condition included in this study that was both physically measured and self-reported in the questionnaire. The prevalence of physically measured diabetes was more than double that of self-reported diabetes (11.7% versus 4.4%, respectively). This indicates that self-reported diabetes is most likely underreported. When combining the responses of the measured and self-reported diabetes, the composite prevalence was 9.1%. All diseases were more prevalent in females compared to males.

**Table 4. Prevalence of single disease conditions by sex and method of measurement in South Africa for 2016 (weighted data).**

| Disease condition | Total % (95% CI) | Male % (95% CI) | Female % (95% CI) |
|---|---|---|---|
| **SELF-REPORTED** | | | |
| Diabetes | 4.4 (3.9–5.0) | 3.7 (3.0–4.5) | 5.1 (4.4–5.9) |
| Bronchitis/COPD | 1.4 (1.1–1.8) | 1.1 (0.8–1.6) | 1.7 (1.3–2.2) |
| Heart disease | 3.1 (2.7–3.5) | 2.3 (1.8–2.9) | 3.8 (3.3–4.5) |
| Cholesterol | 3.5 (2.9–4.2) | 3.0 (2.3–3.8) | 4.1 (3.4–4.9) |
| Stroke | 1.4 (1.1–1.7) | 1.0 (0.7–1.5) | 1.7 (1.4–2.1) |
| TB in last 12 months | 1.2 (0.9–1.5) | 0.9 (0.6–1.3) | 1.5 (1.0–2.0) |
| **MEASURED (BIOMARKER)** | | | |
| HIV | 19.6 (18.2–21.1) | 13.7 (11.8–15.8) | 24.5 (22.7–26.4) |
| Hypertension | 45.0 (43.1–46.9) | 44.1 (41.5–46.7) | 45.8 (43.7–48.0) |
| Anaemia | 24.7 (23.2–26.3) | 16.8 (15.0–18.8) | 31.3 (29.2–33.5) |
| Diabetes (HbA1c) | 11.7 (10.7–12.8) | 9.1 (7.8–10.6) | 13.9 (12.6–15.4) |
| **MEASURED (BIOMARKER) AND SELF-REPORTED** | | | |
| Diabetes (self-report or HbA1c) | 9.1 (8.4–9.9) | 6.9 (5.9–7.9) | 11.1 (10.2–12.3) |

Note: Biomarker prevalence differed slightly from the DHS report due to the different data cleaning methods and cut-offs employed.

The number of diseases present in one individual ranged from zero to six. The difference in the prevalence in the number of diseases by sex was statistically significant (p<0.001). About 49% of the participants had none of the diseases included in the study, with more males compared to females being "disease-free" (55.8% versus 41.8%, respectively, p<0.001) (Table 5). Less than a third of participants (30.8%) had one disease and there was a difference between males and females (29.4% versus 32.1%, respectively, p = 0.0183). Multimorbidity was present in 21% of participants. The prevalence of multimorbidity in females was almost double that of males (26.2% vs 14.8%, respectively) and the difference between the sexes was statistically significant (p<0.001).

Multimorbidity prevalence increased with increasing age in both males and females (Fig 1, S3 Table in S1 File). The prevalence of multimorbidity was consistently higher in females compared to males across the different age groups. Multimorbidity was present at lower levels: 3% for adolescents aged 15–19 years and 10% for 20–29 years old. In females, multimorbidity peaked at 47% in the 60–69 years old; whereas in males, it peaked at 40% in the 70–79 years old. Multimorbidity prevalence dropped slightly in the age group 80 years and over. However, the observed drop is most likely due to uncertainty introduced by a smaller number of people aged 80+ being included in the sample. People with multimorbidity tended to have an older median age compared to those with no multimorbidity (Fig 2), but this difference was not statistically significant.

**Table 5. Number of diseases in individuals by sex in South Africa for 2016 (weighted data).**

| Number of diseases | Total % (95% CI) | Male % (95% CI) | Female % (95% CI) |
|---|---|---|---|
| No disease | 48.6 (47.0–50.1) | 55.8 (53.5–58.1) | 41.8 (40.0–43.4) |
| 1 disease | 30.8 (29.5–32.0) | 29.4 (27.5–31.2) | 32.1 (30.6–33.7) |
| 2 diseases | 14.1 (13.2–15.1) | 10.5 (9.4–11.8) | 17.4 (16.2–18.7) |
| 3 diseases | 5.2 (4.7–5.9) | 3.5 (2.8–4.3) | 6.8 (6.0–7.8) |
| 4 diseases | 1.1 (0.8–1.3) | 0.6 (0.4–0.9) | 1.4 (1.1–1.9) |
| 5 diseases | 0.2 (0.1–0.4) | 0.1 (0.1–0.3) | 0.4 (0.2–0.6) |
| 6 diseases | 0.07 (0.02–0.19) | 0.01 (0.01–0.04) | 0.01 (0.01–0.02) |
| **Multimorbidity ($\geq$ 2 diseases)** | **20.7 (19.5–21.9)** | **14.8 (13.4–16.3)** | **26.2 (24.7–27.7)** |

## Factors associated with multimorbidity

The factors associated with multimorbidity were investigated through a logistic regression (Table 6). For the final model, outliers were dropped and the model was refitted due to its limited ability in predicting multimorbidity in young women with a low BMI (S1 Fig in S1 File). An interaction term between age and sex was added to the model to improve its fitness.

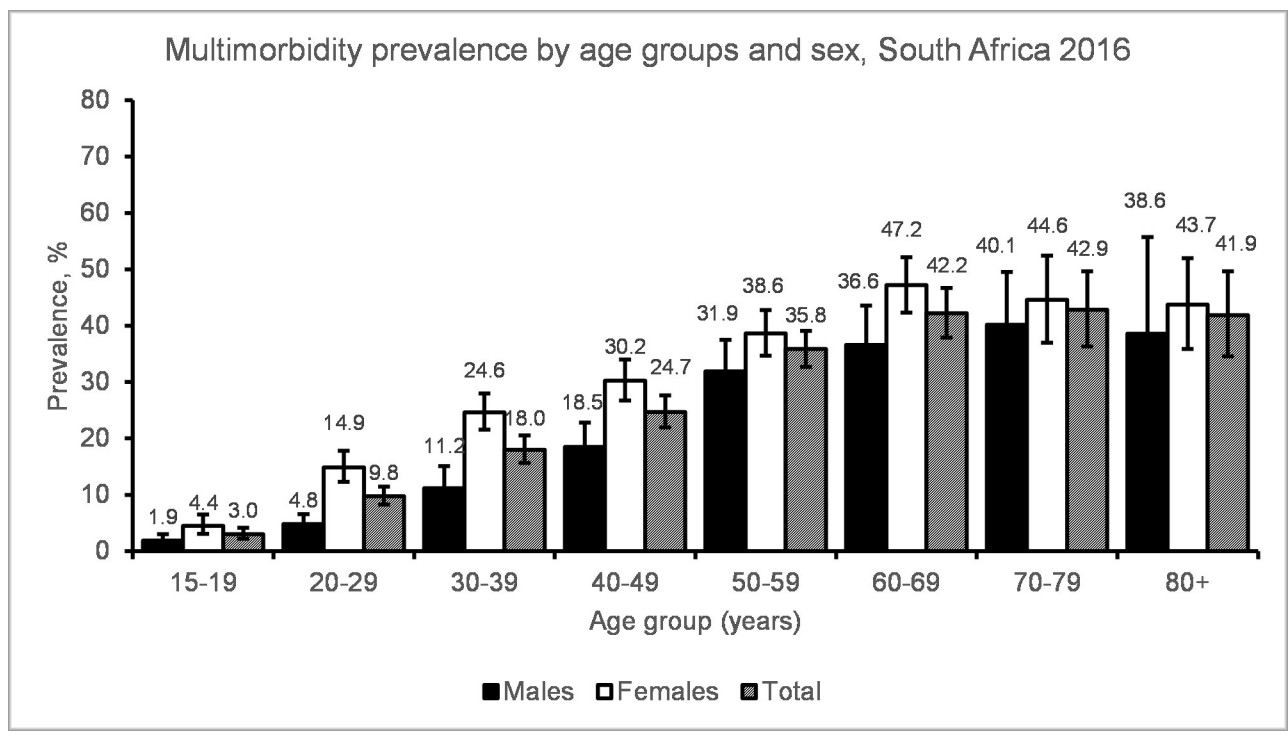

**Fig 1. Estimated multimorbidity prevalence by age group and sex in South Africa in 2016.**

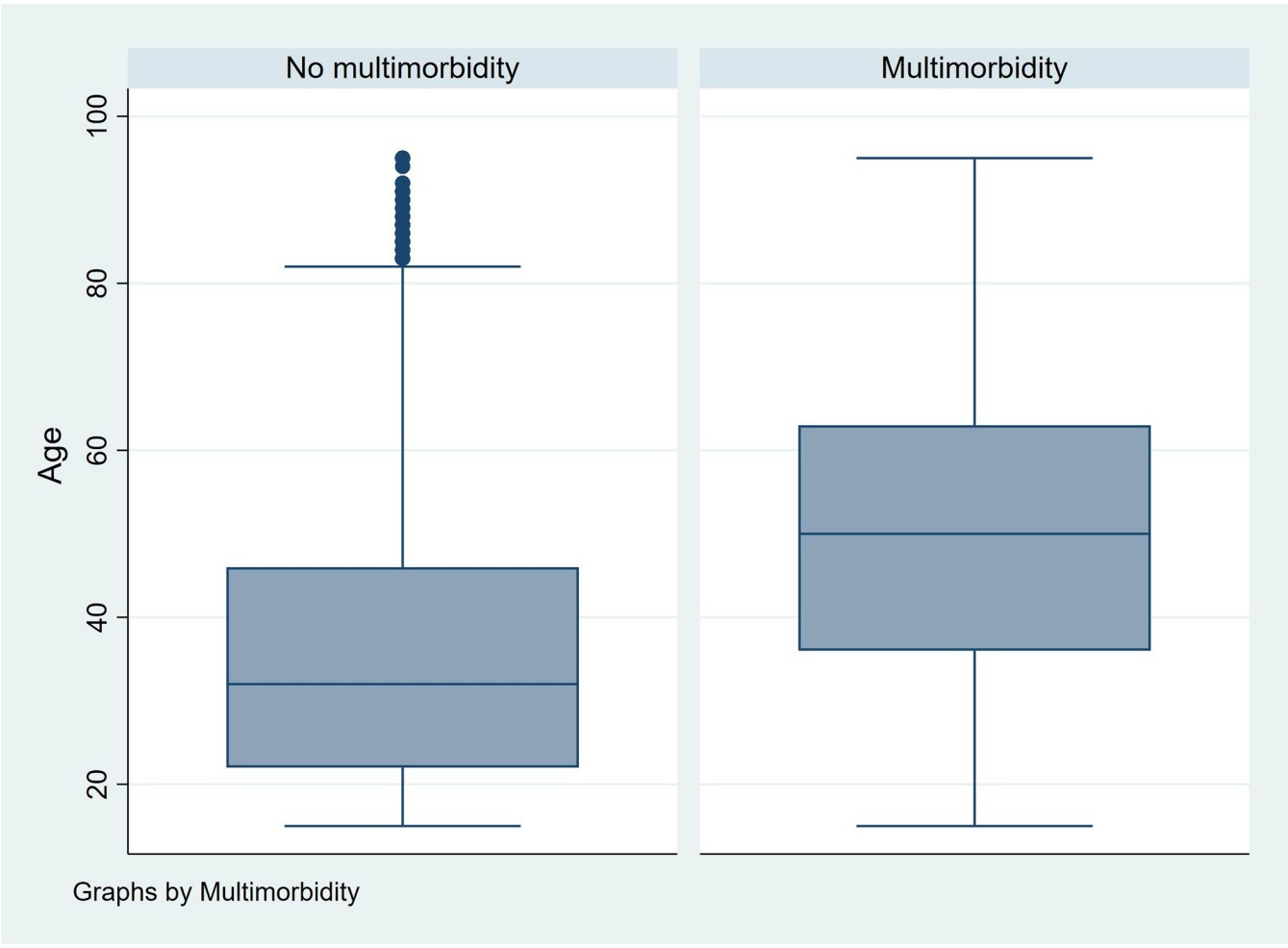

**Fig 2. Multimorbidity status by age.**

The final model showed that the odds of being multimorbid increased with age, with the highest odds if the participant was in the 55–64 years of age group (OR: 24.910, 95% CI: 14.901–41.641, p<0.001), compared to 15–24 years old. Younger females (15–34 years) had larger odds of being multimorbid compared to males in the same age groups.

The odds of being multimorbid were reduced if an individual had tertiary education (OR: 0.722, 95% CI: 0.537–0.970, p = 0.031) compared to only having completed primary school education. Those that were employed had reduced odds of multimorbidity compared to those that were unemployed (OR: 0.813, 95% CI: 0.675–0.979, p = 0.029). Those that had used alcohol recently also reported lowered odds compared to those that were not using alcohol (OR: 0.815, 95% CI: 0.686–0.968, p = 0.02). BMI and current tobacco use were not significant when adjusted for age, sex and other variables. The wealth index was not a predictor of multimorbidity. Additional models can be found in S5 Table in S1 File.

## Discussion

Using the DHS national survey, it was found that one in five South Africans aged 15 years or above was multimorbid. The prevalence of multimorbidity generally increased with age and reached 42% in the 60 years and older age groups. The prevalence of multimorbidity was

**Table 6. Factors associated with multimorbidity.**

| Variable | Unadjusted Odds ratios (95% CI) | Final model (Model 3) Odds ratio (95% CI) |
|---|---|---|
| Age category *(Reference: 15–24 year)* | | |
| • 25–34 years | **2.982 (2.407–3.695)** * | **3.923 (2.299–6.695)** * |
| • 35–44 years | **4.861 (3.769–6.269)** * | **8.417 (5.101–13.890)** * |
| • 45–54 years | **7.527 (5.844–9.694)** * | **14.165 (8.654–23.185)** * |
| • 55–64 years | **11.764 (8.837–15.662)** * | **24.910 (14.901–41.641)** * |
| • 65+ years | **14.181 (10.951–18.364)** * | **23.062 (13.719–38.766)** * |
| Sex *(Reference: Male)* | **2.038 (1.804–2.301)** * | 1.135 (0.831–1.551) |
| Age category and sex interaction | | |
| • 15–24#Female | - | **2.734 (1.498–4.988)** * |
| • 25–34#Female | - | **1.896 (1.169–3.075)** * |
| • 35–44#Female | - | 1.340 (0.842–2.132) |
| • 45–54#Female | - | 1.089 (0.676–1.755) |
| • 55–64#Female | - | 0.866 (0.558–1.345) |
| • 65+#Female | - | **1 (omitted)** |
| Urban *(Reference: Rural)* | **0.817 (0.721–0.925)** * | 1.107 (0.901–1.360) |
| Education *(Reference: Primary)* | | |
| • Secondary | **0.423 (0.372–0.480)** * | 0.966 (0.819–1.140) |
| • Tertiary | **0.323 (0.251–0.414)** * | **0.722 (0.537–0.970)*** |
| Wealth index (Reference: Poorest) | | |
| • Poorer | 0.995 (0.829–1.194) | 1.067 (0.864–1.317) |
| • Middle | 1.076 (0.874–1.324) | 1.126 (0.867–1.464) |
| • Richer | 1.036 (0.845–1.270) | 1.034 (0.778–1.374) |
| • Richest | 0.901 (0.713–1.138) | 0.754 (0.545–1.044) |
| Employed *(Reference: Not employed)* | **0.744 (0.643–0.861)** * | **0.813 (0.675–0.979)*** |
| BMI *(Reference: Underweight)* | | |
| • Normal weight | 0.961 (0.679–1.361) | 0.892 (0.609–1.309) |
| • Overweight | **1.779 (1.227–2.581)** * | 1.033 (0.680–1.571) |
| • Obesity group 1 | **2.536 (1.759–3.655)** * | 1.213 (0.793–1.854) |
| • Obesity group 2 | **2.94 (1.965–4.397)** * | 1.340 (0.840–2.137) |
| • Obesity group 3 | **3.518 (2.367–5.228)** * | 1.527 (0.956–2.438) |
| Current alcohol use *(Reference: No current alcohol use)* | **0.571 (0.498–0.653)** * | **0.815 (0.686–0.968)** * |
| Current tobacco use *(Reference: No current tobacco use)* | **0.704 (0.592–0.838)** * | 0.893 (0.710–1.122) |

higher in females compared to males, but the difference was larger in younger age groups. Our study corroborates other studies that have found high levels of chronic diseases in the sub-Saharan region. For example, an analysis of DHS surveys in 33 sub-Saharan African countries (excluding South Africa), found that there was a high prevalence of hypertension, anaemia, underweight, overweight and obesity in females 15 years or above [45].

Several other national surveys have been analysed to determine the prevalence of multimorbidity in South Africa [19–21, 46]. The prevalence estimates varied from 2.8% [21] to 63.4% [20], although these studies looked at differing age groups, used varying data collection methods and included different disease conditions. The 2003 World Health Survey which surveyed adults 18 years older found a standardised prevalence of 11.2% [19]. Two waves of the National Income Dynamic Surveys (2008 and 2012) found a low prevalence of 2.7% and 2.8%, respectively [21]. The same 2008 dataset was analysed using different methods but found a similar

low prevalence of 4% [46]. Garin *et al.* [20] used the 2007/2008 World Health Organization Study on global AGEing (SAGE) and adult health and found a prevalence of 63.4% in adults over the age of 50 years. Most of these studies included self-reported diseases and physically measured hypertension. Self-reported diseases are likely to be underreported as people may be unaware that they have a disease. The DHS physically measured more diseases compared to the National Income Dynamic Surveys and World Health Survey (i.e. HIV, diabetes and anaemia) which may explain its ability to detect higher levels of multimorbidity. The number of disease conditions included in each study also varied. The Garin *et al.* [20] study was restricted to adults over the age of 50 years and included a larger number of disease conditions (e.g. depression, cognitive impairment, edentulism and obesity as a disease condition) and therefore reported higher prevalence of multimorbidity. A recent analysis of Wave 2 SAGE (2014/2015) [47] of adults aged 45 years and older, included fewer disease conditions than Garin *et al.* [20] (7 vs. 12); reported a multimorbidity prevalence of 21%. Another discrepancy to note is that the 2016 DHS was more recently conducted than the other national surveys.

In terms of factors associated with multimorbidity, an increasing age was associated with being multimorbid. This follows trends in reporting in international [39] and the South African literature on multimorbidity. Multimorbidity is often associated with older adults, especially in high income countries [48] due to shifting demographic trends whereby people are living longer, ageing and developing chronic diseases of lifestyle. However, our study had an interesting finding in observing that multimorbidity was present in 10% of young adults between the ages of 20–29 years. This is most likely attributed to the high prevalence of single disease such as HIV, anaemia and hypertension in South Africa. HIV is known to affect younger adults in South Africa. A South African national HIV prevalence survey indicated that 7.9% of people (4.8% of males, 10.9% of females) aged 15–24 years were HIV positive in 2017 [49]. Also, the prevalence of hypertension in South Africa is thought to be increasing. In young South Africans, hypertension is frequently associated with having a family history of the disease (suggesting a genetic component) and obesity or metabolic syndrome [50, 51]. In this study, approximately 32% of people with HIV under 30 years of age, also had hypertension (S3 Fig in S1 File). This has implications for young people in that they will have to be on lifelong treatment for both diseases.

The present study showed that having tertiary education decreased the odds of multimorbidity, this has been noted both locally and internationally [19, 20, 52]. However, a systematic review of education levels and multimorbidity in Southeast Asia found the association was inconsistent [53]. This study found that being employed decreased the odds of multimorbidity. Similar results were found in an analysis of social determinants and multimorbidity in South Africa [46]. Yet, this could also be interpreted to mean that healthier people are more likely to be employed. In a systematic review of multimorbidity and its impact on workers, multimorbidity was found to have a negative impact on work, worsening absenteeism and lowering employability [54]. The wealth index was not significantly associated with multimorbidity. The relationship between wealth and multimorbidity in this study may be unclear as the diseases included may have different patterns according to the individual disease. For example, HIV could be associated with being in a lower wealth quintile, while cardiovascular diseases such as diabetes could be associated with being in a higher wealth quintile. The same argument could be used to explain the findings on BMI. This study indicated that having a high BMI could be associated with multimorbidity but the findings were not significant. High BMI has been identified as associated with multimorbidity in other studies [55]. However, the inclusion of HIV and anaemia could mean that people with lower BMIs were also prone to being multimorbid. An interesting finding was that alcohol use was associated with decreased odds of multimorbidity. A study of binge drinking among adults in the United States found that binge

drinkers tended to have lower levels of multimorbidity [56]. They related these findings to the 'sick quitter' hypothesis whereby adults stop drinking due to interactions with prescribed medications [57].

## Limitations

The current analysis was limited to the data available and disease conditions asked about in the original survey. Additional disease conditions (e.g. cancer) could have been included in the analysis, but the survey in question only asked if the individual had "ever" had the disease. A strength of this study is that disease conditions were only included if the person could have been considered to have the disease at the time of the survey or at a time close to when the study took place. Many studies of multimorbidity include past and current disease conditions without distinction. Had there been included more disease conditions, the prevalence estimates would have most likely be higher. Also, we did not account for pregnancy status in our calculation of BMI.

The study is limited to a simple count of diseases to determine multimorbidity. Studies done with electronic health records or surveyed people specifically for multimorbidity may have taken the severity of diseases into account. Nonetheless, the DHS provides a robust source of data that could be analysed in other LMICs to generate information about multimorbidity where little is still known.

The analysis included self-reported and measured (biomarker) diseases. Self-reported diseases may have been underreported due to participants being unaware that they have a disease. In this study, the prevalence of measured (biomarker) diseases was higher than self-reported diseases. In addition, this study was cross-sectional by nature meaning that we cannot confer temporality.

## Conclusion

This study showed that one in five South Africans, 15 years or above, are managing more than one disease condition. Multimorbidity started in adolescents and increased with age. Females were more frequently affected than males. It was found that having tertiary education and being employed lowered the odds of multimorbidity.

The high prevalence of multimorbidity needs to be addressed in South Africa. This could be done in a twofold manner: (a) by reducing the high prevalence of single diseases such as hypertension and (b) by simultaneously targeting people with existing diseases to reduce their chances of becoming multimorbid. More studies are needed to identify common disease clusters to assist in the endeavour of targeting high risk people. More studies are also needed to determine whether the trends in multimorbidity are changing in the country. For example, to understand whether policies aimed at diseases such as HIV and hypertension have helped to decrease multimorbidity. Also, information is needed on how emerging diseases such as COVID-19 may affect people with multimorbidity in South Africa.

## Supporting information

**S1 File.**
(DOCX)

## Acknowledgments

We would like to acknowledge the DHS programme for access to the dataset. We would also like to acknowledge Dr Oluwatoyin Awotiwon and Prof Ali Dhansay for their assistance with the inclusion of disease conditions.

## Author Contributions

**Conceptualization:** Rifqah Abeeda Roomaney, Brian van Wyk, Annibale Cois, Victoria Pillay-van Wyk.

**Formal analysis:** Rifqah Abeeda Roomaney, Annibale Cois.

**Investigation:** Rifqah Abeeda Roomaney, Brian van Wyk, Annibale Cois, Victoria Pillay-van Wyk.

**Methodology:** Rifqah Abeeda Roomaney, Brian van Wyk, Annibale Cois, Victoria Pillay-van Wyk.

**Resources:** Brian van Wyk, Victoria Pillay-van Wyk.

**Software:** Rifqah Abeeda Roomaney, Annibale Cois.

**Supervision:** Brian van Wyk, Annibale Cois, Victoria Pillay-van Wyk.

**Writing – original draft:** Rifqah Abeeda Roomaney.

**Writing – review & editing:** Rifqah Abeeda Roomaney, Brian van Wyk, Annibale Cois, Victoria Pillay-van Wyk.

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
