## [Decision Letter · Decision Letter 0]

18 Feb 2022

PONE-D-21-34287One in five South Africans are multimorbid: An analysis of the 2016 Demographic and Health SurveyPLOS ONE

Dear Dr. Roomaney,

Thank you for submitting your manuscript to PLOS ONE. After careful consideration, we feel that it has merit but does not fully meet PLOS ONE’s publication criteria as it currently stands. Therefore, we invite you to submit a revised version of the manuscript that addresses the points raised during the review process.

We look forward to receiving your revised manuscript.

Kind regards,

Mona Pathak, PhD

Academic Editor

PLOS ONE

Journal Requirements:

Reviewers' comments:

Reviewer's Responses to Questions

**Comments to the Author**

1. Is the manuscript technically sound, and do the data support the conclusions?

Reviewer #1: Yes

Reviewer #2: Yes

2. Has the statistical analysis been performed appropriately and rigorously? 

Reviewer #1: Yes

Reviewer #2: Yes

3. Have the authors made all data underlying the findings in their manuscript fully available?

Reviewer #1: Yes

Reviewer #2: No

4. Is the manuscript presented in an intelligible fashion and written in standard English?

Reviewer #1: Yes

Reviewer #2: Yes

5. Review Comments to the Author

Reviewer #1: The manuscript is

- technically sound and contributing some information to the global scientific audience,

-used appropriate statistical analysis,

-presented relevant data, and

-written in standard English

All the comments from reviewer 1 are available in comment section of attached file.

Reviewer #2: The issue on multimorbidity is an important health challenge particularly in sub-Saharan Africa due to the associated consequences coupled with the sub-region’s weak health system. Therefore, I commend the authors for researching on the issue. The manuscript is well written and can be accepted for possible publication after addressing the following comments:

1. The statement of the problem is not too solid. The authors failed to point out clearly the actual limitations of existing studies. What is the actual contribution of the present study to the literature? Is it the use of the national survey?

2. It is also important to let the reader know about few examples of existing studies on multimorbidity: highlighting on their focus and associated factors of multimorbidity.

3. The paragraphs on DHS in the introduction section (Lines 68 -80) should be moved to the methods section.

4. The literature on the discussion of multimorbidity prevalence is quite lengthy. However, the authors did not relate the finding to other sub-Saharan African countries, particularly studies that have used the DHS. While comparison with studies like SAGE and World Health Survey is good, the authors have to be mindful of the years in which these surveys were conducted.

5. The authors failed to discuss some important findings of the study including the association between educational attainment, employment status and current alcohol use and multimorbidity. It is important to relate the findings to the literature and provide possible explanation as well.

6. The methods section needs to be re-written and restructured for easy understanding. For example, under description of included variables, it is important to let the reader know what the outcome variable is, before explaining how it was measured. It is confusing to read paragraphs of sentences before knowing about the outcome variable. Lines 168- 180 should come immediately after the title ‘description of included variables’. Lines 182-187 should be moved to ‘Analysis’. Likewise, lines 190 -194 should be moved to ‘Analysis’ and Lines 202 – 213 should proceed line 194

7. Line 48 – delete through preceding having

8. Lines 195-201 should also be summarized under ‘other variables of interest’

9. What is the justification for measuring diabetes using both self reported and physical measurement? Why didn’t you adopt the same procedure for other diseases such as hypertension?

10. Line 121- rephrase the sentence “self-reported diabetes information was combined for participants without biomarker data” for clarity

11. Line 156, explain what coding the variables as binary implies. Ever or never? Yes or no?

12. Line 157- regarding the measurement of BMI, was the BMI of pregnant women taken? If yes why, if no, why not?

13. Line 171-reference the sentence “Various studies have used this technique when doing secondary data analysis”.

14. Line 216- the use of “over” implies individuals 15 years were excluded from the study. Rephrase the sentence.

15. Line 220- insert had before completed

16. Line 222- delete “and” before did

17. Line 223- clarify the sentence “there were significant difference in participation between males and females by province”. Participation in what?

18. Line 277-delete the sentence

19. Line 300- the findings indicate that it peaked at aged 55-64 years. Hence kindly rephrase the sentence

20. Line 334-335: “In our study, approximately 44% of people with HIV under 30 years of age, also had hypertension”. Which Table or figure indicates it.

6. PLOS authors have the option to publish the peer review history of their article (what does this mean?). If published, this will include your full peer review and any attached files.

Reviewer #1: **Yes: **Dr. Kassa Demissie Abdi (PhD)

Reviewer #2: No

---

## [Author Response · Author response to Decision Letter 0]

25 Feb 2022

<Please see attached Ms Word file for the proper format.>

Manuscript ID PONE-D-21-34287

Response to Reviewers

Dear Dr Pathak,

Thank you for providing us with the opportunity to submit a revised draft of the manuscript “One in five South Africans are multimorbid: An analysis of the 2016 Demographic and Health Survey” to PLOS ONE. We appreciate the time and effort that you and the reviewers have given us, and the chance to revise, improve and strengthen our manuscript. 

Additional text was added to the Introduction and Discussion sections. The Methods section was restructured as Reviewer 2 suggested. Various tracked changes made by Dr Abdi were accepted. We have marked the changes within the manuscript using the tracked change function and insertions were marked in blue highlight. Please see our responses to reviewer comments below.

Comment Response Page & line number

Editor comments (Dr Mona Pathak)

1. Please ensure that your manuscript meets PLOS ONE's style requirements, including those for file naming. Document was checked and meets the standards described in the attachments. N/A.

2. Please provide additional details regarding participant consent. In the ethics statement in the Methods and online submission information, please ensure that you have specified what type you obtained (for instance, written or verbal, and if verbal, how it was documented and witnessed). 

If your study included minors, state whether you obtained consent from parents or guardians. If the need for consent was waived by the ethics committee, please include this information. Additional information was added to the Methods section and the section on Patient consent for publication:

“All participants signed consent forms to participate in the SADHS 2016. For this secondary data analysis, the anonymised dataset with necessary permissions was obtained from the DHS programme. In addition, ethics clearance was granted by the Biomedical Science Research Ethics Committee of the University of the Western Cape (BM20/5/8) as part of the lead author’s PhD project.”

 Page 6, Line 118 – 122.

3. Please review your reference list to ensure that it is complete and correct. 

If you have cited papers that have been retracted, please include the rationale for doing so in the manuscript text, or remove these references and replace them with relevant current references. If you need to cite a retracted article, indicate the article’s retracted status in the References list and also include a citation and full reference for the retraction notice.

Any changes to the reference list should be mentioned in the rebuttal letter that accompanies your revised manuscript. New references were added as indicated in the response to reviewers (below).

References were checked for completeness and errors. No errors were found.

No retracted papers are cited. N/A.

Reviewer 1 (Dr Kassa Demissie Abdi)

4. Reviewer #1: The manuscript is

- technically sound and contributing some information to the global scientific audience,

-used appropriate statistical analysis,

-presented relevant data, and

-written in standard English We thank Dr Kassa Demissie Abdi for the very helpful review, edits and comments made. N/A.

5. How data on the age, sex, residences, educational level and employment status of participants were synthesized from SADHS (2016)? Done. Details on age, sex, locality and employment status were added to the Methods section.

“The ages of participants were taken from DHS 2016 dataset. Participants under the age of 15 years were excluded. Where appropriate, age was analysed in 10-year age bands. The variable sex was included and participants were coded as male or female. Locality was included and coded as either urban or rural. Educational attainment was also included and described by the ‘highest grade or form you completed at that level’. This study divided the responses into three categories: primary school or less, secondary school, and tertiary education. Employment status was coded as employed (currently working) or unemployed.” Page 10, Line 189 – 195.

6. Better to use ≥ 6.5 mmol instead of >=6.5 mmol Done. Symbol inserted. Page 8, Line 151.

7. The results of the first ELISA was included in this study. Done. Wording changed to suggested wording. Page 9, Line 157-158.

8. Participants were then categorized either having no anaemia or having anaemia. The degree of anaemia was characterized as mild, moderate or severe. Done. Wording changed to suggested wording. Page 9, Line 164 – 166.

9. The first measurement was excluded and the average of the remaining repeated measurements were taken. Done. Wording changed to suggested wording. Page 9, Line 169 – 170.

10. Further details of data collection, cleaning and coding is listed in S2 Table Done. Wording changed to suggested wording. Page 9, Line 178-179.

11. Is it to mean variables? Done. Corrected to “Variables”. Page 15, Line 281.

12. Is it medically sound to combine them? Blood was only taken for a sub-sample of the participants - 6 763 participants provided HbA1c measurements whereas 10 292 participants provided information on self-reported diabetes status.

Instead of setting the diabetes information as missing for 3 259 of the participants that did not have HbA1c information, we combined the results with the self-reported diabetes data. We believe it was appropriate for a study of multimorbidity where it would be important to reduce the amount of missing information in the sample.

 N/A.

13. Used. Alcohol use would have been expected to predict multimorbidity. Please, check for collinearity or confounding? We checked our model for collinearity and no problems were detected. 

The model was adjusted for a series of important potential confounders, however, residual confounding is possible. Extra text was added to the Discussion to explain the concept of the ‘sick quitter’ hypothesis:

“An interesting finding was that alcohol use was associated with decreased odds of multimorbidity. A study of binge drinking among adults in the United States found that binge drinkers tended to have lower levels of multimorbidity [56]. They related these findings to the ‘sick quitter’ hypothesis whereby adults stop drinking due to interactions with prescribed medications [57].” Page 24, Line 419 – 423.

14. Make it clear i.e. higher or lower? Done. Sentence changed to:

“The prevalence of multimorbidity was higher in females compared to males, but the difference was larger in younger age groups” Page 22, Line 360.

Reviewer 2

15. The issue on multimorbidity is an important health challenge particularly in sub-Saharan Africa due to the associated consequences coupled with the sub-region’s weak health system. Therefore, I commend the authors for researching on the issue. The manuscript is well written and can be accepted for possible publication after addressing the following comments: We thank the reviewer for the positive comments. N/A.

16. The statement of the problem is not too solid. 

The authors failed to point out clearly the actual limitations of existing studies. 

What is the actual contribution of the present study to the literature? Is it the use of the national survey?

17. It is also important to let the reader know about few examples of existing studies on multimorbidity: highlighting on their focus and associated factors of multimorbidity. We thank the reviewer for the suggestions. We inserted a few sentences that both a) mentions existing studies and their problems b) more clearly states why we did the analysis and what the contribution of the study is.

“While multimorbidity research has been emerging in the country for the past decade, few studies have reported the prevalence of multimorbidity and factors associated with it in a consistent and comparable manner [18]. The authors conducted a systematic review of multimorbidity prevalence studies in South Africa and found significant heterogeneity in the study designs as well as the estimates of prevalence [18]. Of the studies included [19-27], the prevalence of multimorbidity ranged from 3 to 87%. In addition, the factors associated with multimorbidity were disparate and at times contradictory. Among the factors that were occasionally associated with multimorbidity in South Africa were: age, being female, locality, education level, body mass index (BMI) and marital status.” 

“Prevalence estimates form an important part of the information used for evidence-based health decision-making. Given the lack of studies conducted about multimorbidity in South Africa, we aimed to determine the prevalence of multimorbidity by age group and sex in the country using the 2016 Demographic and Health Surveys (SADHS 2016). In addition, this paper reports the process and results derived from a systematic analysis of the SADHS 2016 to establish factors associated with multimorbidity in the South African population. The SADHS 2016 is unique in South Africa in that it is a nationally representative survey which includes biomarkers for the measurement of HIV, HbA1c (diabetes), blood pressure and anaemia status.” Page 2, Line 65 – 71.

Page 2, Line 73-81

18. The paragraphs on DHS in the introduction section (Lines 68 -80) should be moved to the methods section. Done. Paragraph was moved to Methods section. Page 4, Line 84 – 94.

19. The literature on the discussion of multimorbidity prevalence is quite lengthy. However, the authors did not relate the finding to other sub-Saharan African countries, particularly studies that have used the DHS. 

While comparison with studies like SAGE and World Health Survey is good, the authors have to be mindful of the years in which these surveys were conducted. Done. In addition, we included the following:

 “Our study corroborates other studies that have found high levels of chronic diseases in the sub-Saharan region. For example, an analysis of DHS surveys in 33 sub-Saharan African countries (excluding South Africa), found that there was a high prevalence of hypertension, anaemia, underweight, overweight and obesity [45].”

We noted that the years of the surveys included differed.

“Another discrepancy to note is that the 2016 DHS was more recently conducted than the other national surveys.” Page 22, Line 361 – 364.

Page 22, Line 386.

20. The authors failed to discuss some important findings of the study including the association between educational attainment, employment status and current alcohol use and multimorbidity. It is important to relate the findings to the literature and provide possible explanation as well. This section was expanded upon:

“The present study showed that having tertiary education decreased the odds of multimorbidity, this has been noted both locally and internationally [19,20,52]. However, a systematic review of education levels and multimorbidity in Southeast Asia found the association was inconsistent [53]. This study found that being employed decreased the odds of multimorbidity. Similar results were found in an analysis of social determinants and multimorbidity in South Africa [46]. Yet, this could also be interpreted to mean that healthier people are more likely to be employed. In a systematic review of multimorbidity and its impact on workers, multimorbidity was found to have a negative impact on work, worsening absenteeism and lowering employability [54]. The wealth index was not significantly associated with multimorbidity. The relationship between wealth and multimorbidity in this study may be unclear as the diseases included may have different patterns according to the individual disease. For example, HIV could be associated with being in a lower wealth quintile, while cardiovascular diseases such as diabetes could be associated with being in a higher wealth quintile. The same argument could be used to explain the findings on BMI. This study indicated that having a high BMI could be associated with multimorbidity but the findings were not significant. High BMI has been identified as associated with multimorbidity in other studies [55]. However, the inclusion of HIV and anaemia could mean that people with lower BMIs were also prone to being multimorbid. An interesting finding was that alcohol use was associated with decreased odds of multimorbidity. A study of binge drinking among adults in the United States found that binge drinkers tended to have lower levels of multimorbidity. They related these findings to the ‘sick quitter’ hypothesis whereby adults stop drinking due to interactions with prescribed medications [56].”

Page 24, Line 402 - 423

21. The methods section needs to be re-written and restructured for easy understanding. For example, under description of included variables, it is important to let the reader know what the outcome variable is, before explaining how it was measured. It is confusing to read paragraphs of sentences before knowing about the outcome variable. 

Lines 168- 180 should come immediately after the title ‘description of included variables’. 

Lines 182-187 should be moved to ‘Analysis’. Likewise, lines 190 -194 should be moved to ‘Analysis’ and Lines 202 – 213 should proceed line 194

 Done. We thank the reviewer for the excellent suggestions. The text has been restructured as the reviewer suggested. Methods section.

22. Line 48 – delete through preceding having Done. Page 3, Line 46.

23. Lines 195-201 should also be summarized under ‘other variables of interest’ Done. Paragraph moved. Page 10, Line 181

24. What is the justification for measuring diabetes using both self reported and physical measurement? 

Why didn’t you adopt the same procedure for other diseases such as hypertension? Blood was only taken for a sub-sample of the participants - 6 763 participants provided HbA1c measurements whereas 10 292 participants provided information on self-reported diabetes status.

Instead of setting the diabetes information as missing for 3 259 of the participants that did not have HbA1c information, we combined the results with the self-reported diabetes data. We believe it was appropriate for a study of multimorbidity where it would be important to reduce the amount of missing information in the sample.

The same procedure was not used for hypertension because we did not include self-reported hypertension in our study. Two clinicians independently rated which self-reported diseases were suitable to include in the analysis and they assessed that self-reported hypertension was not suitable due to the way in which the question was asked e.g. “Have you ever been diagnosed with hypertension?”. In their assessment, hypertensive status can be changed through diet and it would be inappropriate to conclude that if a person was once diagnosed with hypertension that they could be assumed to be hypertensive. In addition, the proportion of missing data for the hypertension biomarker was lower than that of HbA1c. N/A.

25. Line 121- rephrase the sentence “self-reported diabetes information was combined for participants without biomarker data” for clarity Done. Sentence changed:

“For participants without HbA1c data, their disease status was based on their self-assessment of whether they had diabetes or not.” Page 8, Line 152 -153.

26. Line 156, explain what coding the variables as binary implies. Ever or never? Yes or no? Done. Text added:

Both variables were coded as binary (e.g., Yes/No).

 Page 11, Line 207.

27. Line 157- regarding the measurement of BMI, was the BMI of pregnant women taken? If yes why, if no, why not? The BMI of pregnant women was not adjusted for. Only 2.3% (n=117) of the women that had BMI measurements were pregnant, and of those 71.8% were normal weight and overweight.

We have added this as a limitation in the Discussion. 

“Also, we did not account for pregnancy status in our calculation of BMI.” Page 25, Line 433.

28. Line 171-reference the sentence “Various studies have used this technique when doing secondary data analysis”. Done. A systematic review of systematic reviews was cited.

Johnston MC, Crilly M, Black C, Prescott GJ, Mercer SW. Defining and measuring multimorbidity: a systematic review of systematic reviews. European journal of public health. 2019 Feb 1;29(1):182-9. Page 6, Line 32.

29. Line 216- the use of “over” implies individuals 15 years were excluded from the study. Rephrase the sentence. Done. Sentence changed to:

“There were 10 336 youth and adults included in the sample…” Page 14, Line 271.

30. Line 220- insert had before completed Done. The word “had” was inserted. Page 14, Line 274

31. Line 222- delete “and” before did Done. Wording changed as suggested. Page 14

32. Line 223- clarify the sentence “there were significant difference in participation between males and females by province”. Participation in what? Done. The wording was changed to clarify that there was a difference in the proportion between males and females by province.

“There were significant differences between the proportion of males and female participants in the sample, by province (p < 0.001) and wealth quintile (p = 0.018)” Page 14, Line 278.

33. Line 277-delete the sentence Unclear. Line 277 was the title for Figure 2. N/A.

34. Line 300- the findings indicate that it peaked at aged 55-64 years. Hence kindly rephrase the sentence Fig.1 shows that multimorbidity prevalence was 42% in the 60 years and older age groups. The sentence was changed to:

“The prevalence of multimorbidity generally increased with age and reached 42% in the 60 years and older age groups…”

In addition, a sentence in the abstract was updated:

“Multimorbidity increased with age; with the highest odds in the 55 - 64 years old age group…” Page 22, Line 359.

Page 2, Line 34.

35. Line 334-335: “In our study, approximately 44% of people with HIV under 30 years of age, also had hypertension”. Which Table or figure indicates it. Thank you for picking this up. We added “data not shown.” Page 24, Line 400.

---

## [Decision Letter · Decision Letter 1]

23 Mar 2022

PONE-D-21-34287R1One in five South Africans are multimorbid: An analysis of the 2016 Demographic and Health SurveyPLOS ONE

Dear Dr. Roomaney,

Thank you for submitting your manuscript to PLOS ONE. After careful consideration, we feel that it has merit and acceptable for publication after some minor comments need to be incorporated in drafting of manuscript. Therefore, we invite you to submit a revised version of the manuscript that addresses the points raised during the review process.

We look forward to receiving your revised manuscript.

Kind regards,

Mona Pathak, PhD

Academic Editor

PLOS ONE

Journal Requirements:

Reviewers' comments:

Reviewer's Responses to Questions

**Comments to the Author**

1. If the authors have adequately addressed your comments raised in a previous round of review and you feel that this manuscript is now acceptable for publication, you may indicate that here to bypass the “Comments to the Author” section, enter your conflict of interest statement in the “Confidential to Editor” section, and submit your "Accept" recommendation.

Reviewer #1: (No Response)

Reviewer #2: (No Response)

2. Is the manuscript technically sound, and do the data support the conclusions?

Reviewer #1: Yes

Reviewer #2: Yes

3. Has the statistical analysis been performed appropriately and rigorously? 

Reviewer #1: Yes

Reviewer #2: Yes

4. Have the authors made all data underlying the findings in their manuscript fully available?

Reviewer #1: Yes

Reviewer #2: Yes

5. Is the manuscript presented in an intelligible fashion and written in standard English?

Reviewer #1: Yes

Reviewer #2: Yes

6. Review Comments to the Author

Reviewer #1: (No Response)

Reviewer #2: All comments raised in the previous review have been addressed by the authors. However, there are some few additional comments to be addressed before publication.

1. Line 246: delete the sub-title factors associated with multimorbidity

2. Line 267: Cut the sentence "Three models were constructed for logistic regression with multimorbidity as the dependent variable and paste it at line 257 before Model 1

3. Line 284: the sentence whiles females had a slightly higher prevalence of TB in the last 12 months, though...... is not complete. Kindly check and complete it.

4. Line 309: delete "disease" after six

5.Line 357: change over the age 15 years to aged 15years or above

6. Line 362-364: which group of people did the study focused on? adult men, women or both?

7. Line 333: if the data is available, kindly show it as an appendix to erase doubts among readers

8.Line 440: change over the age 15 years to 15 years or above

7. PLOS authors have the option to publish the peer review history of their article (what does this mean?). If published, this will include your full peer review and any attached files.

Reviewer #1: **Yes: **Dr. Kassa Demissie Abdi

Reviewer #2: No

---

## [Author Response · Author response to Decision Letter 1]

24 Mar 2022

Manuscript ID PONE-D-21-34287R1

Response to Reviewers

Dear Dr Pathak,

Thank you for once again providing us with the opportunity to submit a revised draft of the manuscript “One in five South Africans are multimorbid: An analysis of the 2016 Demographic and Health Survey” to PLOS ONE. 

We thank Reviewer 2 for the thorough review and for suggesting that data be added to the supplementary appendix. We have added an extra graph to the appendix and applied all the changes the reviewer suggested. Please see our responses to the reviewer comments below.

Comment 

Editor comments (Dr Mona Pathak)

1. Please review your reference list to ensure that it is complete and correct. 

If you have cited papers that have been retracted, please include the rationale for doing so in the manuscript text, or remove these references and replace them with relevant current references. If you need to cite a retracted article, indicate the article’s retracted status in the References list and also include a citation and full reference for the retraction notice.

Any changes to the reference list should be mentioned in the rebuttal letter that accompanies your revised manuscript. 

Reference: References were checked for completeness and errors. 

Reviewer 2

1. Line 246: delete the sub-title factors associated with multimorbidity

Response: Done. The sub-title was deleted. Line 221, Page 12.

2. Line 267: Cut the sentence "Three models were constructed for logistic regression with multimorbidity as the dependent variable and paste it at line 257 before Model 1

Response: Done. The sentence was moved to where the reviewer suggested. Line 227-228, Page 12.

3. Line 284: the sentence whiles females had a slightly higher prevalence of TB in the last 12 months, though...... is not complete. Kindly check and complete it.

Response: Done. The sentence was changed to:

Females had a slightly higher prevalence of TB in the last 12 months compared to males, however, the difference was not statistically significant. Line 256 – 257, Page 14.

4. Line 309: delete "disease" after six 

Response: Done. Word was deleted.

 Line 282, Page 16.

5.Line 357: change over the age 15 years to aged 15years or above

Response: Done. Wording changed as suggested. Line 330-331, Page 20.

6. Line 362-364: which group of people did the study focused on? adult men, women or both?

Response: Added text “in females 15 years or above”. Line 337, Page 21.

7. Line 333: if the data is available, kindly show it as an appendix to erase doubts among readers

Response: We thank the reviewer for insisting on adding the data. Upon relooking at the data, the quoted prevalence was for the unweighted whole population and not for those under 30 years. We have added the corrected weighted prevalence for those under 30 years (32%) and added a supporting figure in the supplementary appendix (Fig S3). 

 Line 373, Page 22. Figure S3 in appendix added.

8.Line 440: change over the age 15 years to 15 years or above 

Response: Done. Wording changed as suggested. Line 420, Page 24.

---

## [Decision Letter · Decision Letter 2]

16 May 2022

One in five South Africans are multimorbid: An analysis of the 2016 Demographic and Health Survey

PONE-D-21-34287R2

Dear Dr. Roomaney,

We’re pleased to inform you that your manuscript has been judged scientifically suitable for publication and will be formally accepted for publication once it meets all outstanding technical requirements.

Kind regards,

Carla Pegoraro

Division Editor

PLOS ONE

Reviewers' comments:

Reviewer's Responses to Questions

**Comments to the Author**

1. If the authors have adequately addressed your comments raised in a previous round of review and you feel that this manuscript is now acceptable for publication, you may indicate that here to bypass the “Comments to the Author” section, enter your conflict of interest statement in the “Confidential to Editor” section, and submit your "Accept" recommendation.

Reviewer #2: All comments have been addressed

2. Is the manuscript technically sound, and do the data support the conclusions?

Reviewer #2: Yes

3. Has the statistical analysis been performed appropriately and rigorously? 

Reviewer #2: Yes

4. Have the authors made all data underlying the findings in their manuscript fully available?

Reviewer #2: Yes

5. Is the manuscript presented in an intelligible fashion and written in standard English?

Reviewer #2: Yes

6. Review Comments to the Author

Reviewer #2: (No Response)

7. PLOS authors have the option to publish the peer review history of their article (what does this mean?). If published, this will include your full peer review and any attached files.

Reviewer #2: No

---

## [Editor Report · Acceptance letter]

18 May 2022

PONE-D-21-34287R2 

One in five South Africans are multimorbid: An analysis of the 2016 Demographic and Health Survey 

Dear Dr. Roomaney:

I'm pleased to inform you that your manuscript has been deemed suitable for publication in PLOS ONE. Congratulations! Your manuscript is now with our production department. 

Kind regards, 

on behalf of

Dr Carla Pegoraro 

Staff Editor

PLOS ONE